# The Application of Statins in the Regeneration of Bone Defects. Systematic Review and Meta-Analysis

**DOI:** 10.3390/ma12182992

**Published:** 2019-09-16

**Authors:** Elisabet Roca-Millan, Beatriz González-Navarro, Keila Izquierdo-Gómez, Antonio Marí-Roig, Enric Jané-Salas, José López-López, Eugenio Velasco-Ortega

**Affiliations:** 1School of Dentistry, Faculty of Medicine and Health Sciences, University of Barcelona, 08907 Barcelona, Spain; 2Oral Health and Masticatory System Group-IDIBELL (Institut d’Investigació Biomèdica de Bellvitge), Barcelona University, 08907 Barcelona, Spain; 3Department of Maxillofacial Surgery, University Hospital of Bellvitge, 08907 Barcelona, Spain; 4School of Dentistry, Faculty of Medicine and Health Sciences, Odontological Hospital University of Barcelona, University of Barcelona, 08907 Barcelona, Spain; 5Faculty of Dentistry, University of Seville, 41009 Seville, Spain

**Keywords:** statins, bone formation, bone regeneration, HMG-CoA reductase inhibitors

## Abstract

This systematic review aims to analyze the effect of the local application of statins in the regeneration of non-periodontal bone defects. A systematic study was conducted with the Pubmed/Medline, Embase, Cochrane Library and Scielo databases for in vivo animal studies published up to and including February 2019. Fifteen articles were included in the analysis. The local application of the drug increased the percentage of new bone formation, bone density, bone healing, bone morphogenetic protein 2, vascular endothelial growth factor, progenitor endothelial cells and osteocalcin. Meta-analyses showed a statistically significant increase in the percentage of new bone formation when animals were treated with local statins, in contrast to the no introduction of filling material or the introduction of polylactic acid, both in an early (4–6 weeks) and in a late period (12 weeks) (mean difference 39.5%, 95% confidence interval: 22.2–56.9, *p* <0.001; and mean difference 43.3%, 95% confidence interval: 33.6–52.9, *p* < 0.001, respectively). Basing on the animal model, the local application of statins promotes the healing of critical bone size defects due to its apparent osteogenic and angiogenic effects. However, given the few studies and their heterogenicity, the results should be taken cautiously, and further pilot studies are necessary, with radiological and histological evaluations to translate these results to humans and establish statins’ effect.

## 1. Introduction

Statins, inhibitors of 3-hydroxy-3-methylglutaryl-coenzyme A (HMG-CoA) reductase, are widely used agents for lowering cholesterol concentrations. These drugs reduce the progression and may induce the regression of atherosclerosis and are associated with a reduction in cardiovascular morbidity and mortality [1,2,3,4,5,6,7].

Although this is their main application, because mevalonate, the product of HMG-CoA reductase’s reaction, is the precursor of many other non-steroidal isoprenoid compounds other than cholesterol; their inhibiting role has pleiotropic effects [1,3,5,6,7]. Those include the anti-inflammatory, immunomodulatory and antimicrobial properties [3,4]. Statins have also been shown to interfere in the process of bone turnover and regeneration due to their action on different types of cells, including osteoblasts, osteoclasts, endothelial cells and mesenchymal stem cells [2,3,4,5,6,7]. For this reason, they have been studied for the treatment of osteoporosis without finding clear conclusions [3,5,6]. The potential benefit of these drugs in bone regeneration has recently been studied [2,3,5,7].

Two key points in the development of new applications of statins are its low cost and its relatively good safety profile [1,5]. Side effects are rare but can be serious, especially liver toxicity, myositis and rhabdomyolysis [1,3,4]. This is one of the reasons why the interest in investigating local delivery strategies is growing [3,7]. Secondly, with oral treatment, much of the drug may be lost during first pass of metabolism, so that higher doses will be needed to be effective in the defect site [3,7]. In addition, local application will facilitate the management of the concentration necessary to take advantage of the antimicrobial effect of statins [3].

Studies analyzing the effect of statins on the regeneration of non-periodontal bone defects are still mostly in the preclinical phase, and evaluate radiological, histological and analytical parameters to assess their impact [8,9,10,11,12,13,14,15,16,17,18,19,20,21,22].

In this context, the present systematic review is based on the PICO question. P: animals with induced bone defects; I: local application of statins; C: compared with no treatment or placebo; O: improvement in bone healing.

## 2. Material and Methods

An electronic search was conducted in the Pubmed, Cochrane and Scielo databases in February 2019. No limitations were established regarding the publication date. The following key terms were applied: “Statins” or “HMG-CoA reductase inhibitors;” or “Lipid-lowering medications” and “Bone formation,” or “Bone regeneration,” or “Bone healing,” or “Bone turnover” or “Bone metabolism”.

The titles and abstracts of all the articles located were read to select in vivo studies in animal models analyzing the effect of a local application of statins on bone defects’ regenerations. Works not written in English or Spanish, and reviews and in vitro experimental studies, were excluded. Those in vivo studies that evaluated the effect of statins on implant osseointegration, or as an adjunct to periodontal treatment, and those in which statins were administered systemically, were also discarded.

The full text of the selected studies was then evaluated. Articles in which the sample presented induced osteoporosis; a bone defect was not created; the defect was filled with bone graft or bone substitutes, such as tricalcium phosphate; and when the application of statins was not within the defect, were excluded too. A follow-up of less than 4 weeks or a sample of less than 15 were also used as exclusion criteria.

Due to the little existent literature on the subject, the animal species, the type and dose of statin administered and the rest of the substances that made up the solution or implant introduced in the defect, were not considered.

The qualities and risks of bias of the animal studies were assessed using the ARRIVE (Animal Research: Reporting in vivo Experiments) guidelines [23] and the SYRCLE’s (Systematic Review Centre for Laboratory Animal Experimentation) tool (First version, Central Animal Laboratory, Radboud University Medical Center, Nijmegen, the Netherlands) [24], developed from the Risk of Bias tool of the Cochrane Collaboration. The review itself was assessed using the Preferred Reporting Items for Systematic Reviews and Meta-Analyses (PRISMA) scale of Moher et al, 2009 [25], with 22 items.

Pooled estimates from the studies were analyzed using a continuous random-effects model meta-analysis. The evaluated variable was changes produced as a percentage of new bone under the following intervention: the local application of simvastatin in the created defect versus no introduction of filling material or introduction of placebo.

Forest plots were produced to graphically represent the differences in outcomes of the percentages of new bone values; *p* < 0.001 was used as the level of significance. Heterogeneity was assessed with an de x^2^ test and a I^2^ test. The OpenMeta [analyst] tool (First version, Brown University, Providence, RI, USA) was employed in the statistical analysis.

## 3. Results

Of the 192 articles initially located in the electronic search, 14 were discarded due to duplication. After reading titles and abstracts, 155 were excluded, as they did not fulfill the selection criteria. The full texts of the 23 remaining articles were read, discarding two in which osteoporosis was induced to animals; one as no bone defect was created; another in which statins were administered transdermally; two as the statins’ matrix was bone graft; one in which the drug was introduced at the same time as tricalcium phosphate; and another one because filling material was injected near the defect, but not inside it.

Following the selection process, this review analyzed a total of 15 in vivo animal studies evaluating the effect of local application of statins in the regeneration of induced bone defects (Figure 1, Figure 2 and Figure 3) (Table 1). The total animal population was 546 (18 dogs, 36 rabbits and 492 rats), distributed between 287 as controls and 315 in intervention groups, 56 of them being in both groups at the same time with different bone defects treated with different filling material [15,16,22].

These articles compare the non-introduction of filling material [9,10,12,15,16,20], and the introduction of phosphate-buffered saline [14,15,16,17,18,19], polylactic acid [8,12], collagen sponges [9,16,21], hyaluronic acid [11], polyethylene glycol [11,18], gelatin hydrogel [13,16,22] or hydroxypropyl methylcellulose [17], with the introduction into the defect area of preparations containing statins [8,22].

The statin used and the dose were different depending on the study, with simvastatin being the most employed [8,9,10,12,13,16,17,19,20], followed by lovastatin [11,14,18,22]. Rosuvastatin [15,21] was used in only two of the articles.

In all the works [8,9,10,11,12,13,14,15,16,17,18,19,20,21,22], the filling material was introduced at the time of the bone defect creation, except for one, in which a parallel experiment was carried out, in which the injections were made at the time of surgery, and at 14, 28 and 42 days [18].

The samples were followed for a variable period of time, between 28 and 112 days, with the mean of the studies being 58.9 days. During this time, between one and five analyzes were performed, with two analyzes being the most frequent [8,22].

The studied parameters can be grouped into radiological, histological and analytical, highlighting the amount of new bone [8,9,10,16,18,19,20,21], bone density [9,10,13,18], bone healing [11,12,13,14,15] and bone morphogenetic protein 2 (BMP-2) [8,10,13,16] as the most studied parameters.

In all the trials in which the percentage of new bone formation was analyzed, statistically significant results were obtained when comparing the groups in which the filling material contained statins with the control group [8,10,16,18]. Only in two of them did the necessary data appear in order to compare them quantitatively [8,16]. Three other studies measured the reossification area [9,19,21], with statistically significant results in two of them [19,21]. In the only work in which the ratio of new bone formation was analyzed, the results were statistically significant [20].

Three [10,13,18] of the four studies [9,10,13,18] that analyzed bone density in the defect area showed significant results two months after local application of statins. Bone healing was statistically better in the intervention groups in the five trials in which it was analyzed, considerably reducing the healing periods compared to controls [11,12,13,14,15].

In relation to BMP-2, all the studies that analyzed it found a higher expression or a greater number of cells that stained positive for BMP-2 after the application of statins [8,10,13,16].

The vascular endothelial growth factor (VEGF) was significantly increased after the local treatment in the two cases in which it was analyzed [10,13].

The two trials that evaluated progenitor endothelial cells (EPCs), either in the peripheral blood [8] or in the perilesional tissue [13], obtained a statistically significant increase in the values after the local application of statins.

In three studies, osteoblast levels were analyzed. In one, density was evaluated [13], obtaining a significant increase in the intervention group; in another, the percentage of surface covered with osteoblasts was measured [17], without finding significant differences between both groups. In the last one, the number of osteoblasts in a given area was counted, obtaining a significantly higher value in the intervention group [19].

As for osteocalcin, a significant increase in its expression was also observed after the use of statins in the two trials where it was studied [10,13].

The biomechanical parameters were analyzed twice, obtaining a significant difference of the maximum strength and work to fracture between the intervention group and the control group in one of the studies [14], and of the relative ultimate stress and the relative extrinsic stiffness in the other [13].

Some parameters were only measured in one of the articles analyzed, obtaining in some cases, statistically significant differences with respect to the control group: bone marrow-derived mesenchymal stem cells (BMSCs) [8], hypoxia-inducible factor 1-alpha (HIF- 1α) [8], osteoprotegerin [10], alkaline phosphatase enzyme [10], receptor activator for nuclear factor κ B ligand (RANKL) [10], width of trabecula [13], capillary density [13], mRNA expression of the angiogenic markers endothelial nitric oxide synthase (eNOS) and stromal cell-derived factor 1 (SDF-1) [13], osteoid volume [17], osteoid thickness [17], double-labeled calcein surface [17], fibroblasts [19], osteoclasts [19] and vessel diameter [19].

Separate meta-analyses were performed to analyze the mean differences of percentages of new bone formed between studies [8,16]. Those two trials evaluated the values of that parameter at different points in time (4–6 and 12 weeks) after the application of local simvastatin in the intervention group (*n* = 26) versus no introduction of filling material or introduction of polylactic acid in the control group (*n* = 26).

The forest plot (Figure 4) shows a percentage of new bone mean difference of 39.5% with a *p*-value < 0.001 (95% CI: 22.2 to 56.9, heterogeneity I^2^ = 98.4%, P < 0.001) at 4–6 weeks after the intervention.

The forest plot (Figure 5) shows a percentage of new bone mean difference of 43.3% and a *p*-value < 0.001 (95% CI: 33.6 to 52.9, heterogeneity I^2^ = 97.7%, P < 0.001) at 12 weeks after the intervention.

Even with those encouraging results, regrettably, much value cannot be given to these meta-analyzes, since they only include two studies that are heterogeneous.

## 4. Discussion

In one of the three studies in which the area of reossification was measured [9], no significant results were obtained after the intervention, although in all those in which the percentage of new bone formed was measured [8,10,16,18], a very similar parameter, all the findings were statistically significant. The reason for no significance could be due to the concentrations of simvastatin used, since it was the only work in which it was said that some of the animals in the intervention group suffered neurological sequelae, so that the dose administered was perhaps too high.

In the same study [9], bone density was analyzed, and no significant increase in density was observed at 2 months, unlike in the rest of the studies [10,13,18]. However, this increase in the density did occur at 30 days, but then the values remained below the values of the control group over time. That could also be correlated with the concentration of statins. Too high of a concentration may not have beneficial effects on bone regeneration; on the contrary, it could harm the normal regeneration process and have adverse effects, such as necrosis, the formation of granulation tissue and inflammatory infiltration, and muscular degeneration [11].

Three papers analyzed the osteoblasts: one in which the density was measured and the results were statistically significant [13]; another in which the numbers of osteoblasts per area were counted also, with statistically significant results [19]; and the last one in which the percentage of trabecular surface covered by osteoblasts was evaluated, wherein the values were higher than in the control group but not enough to be meaningful [17]. That lack of significance could be due to the scarcity of the samples. Larger samples could give us results of greater value.

In all the studies that analyzed bone healing [11,12,13,14,15], BMP-2 [8,10,13,16], VEGF [10,13], EPCs [8,13] and osteocalcin [10,11,12,13], statistically significant results were obtained when the intervention group was compared with the control group. These studies showed differences in the type of simvastatin used, doses and other substances that were part of the filling material. This seems to indicate that a concentration of statins which is not too high to avoid the appearance of adverse effects can be very useful in the regeneration of bone defects, independently of the other components of the implanted substance.

It seems that the local application of statins could statistically improve some of the key parameters in bone formation, such as BMSCs [8], osteoprotegerin [10], alkaline phosphatase enzyme [10], RANKL [10] and osteoclasts [19], although more studies are needed to confirm the role of statins in osteogenesis, since these values were analyzed only once.

In one of the papers, it was concluded that the improvement in angiogenesis induced by statins favors the fracture union, after analyzing capillary density and angiogenic markers, such as eNOS and SDF-1 [13].

Recently, three pilot studies in humans have been published using simvastatin for bone regeneration [26,27,28]. In one of them [26], the healing of maxillary third molar postextraction sockets was compared clinically and radiologically after the application of different preservation materials: deproteinized bovine bone mineral with 10% collagen, poly(D,L-lactide-co-glycolide) with a hydroxyapatite/β-tricalcium phosphate scaffold (PLGA/HA); poly(D,L-lactide-co-glycolide) and hydroxyapatite/β-tricalcium phosphate with a 2% simvastatin scaffold (PLGA/HA/S); and spontaneous healing. The results were not as expected, as there were more failures in sockets filled with PLGA/HA scaffolds with and without simvastatin. Scaffolds with simvastatin showed better results, with less clinical complications than scaffolds without simvastatin. The same conclusion comes from the study by Papadimitriou et al., conducted on 14 New Zealand white rabbits, not included in this review for not meeting criteria, when it suggests that the local application of simvastatin, combined with an appropriate carrier, can promote new bone formation [27].

The two other pilot studies compared the use of β-tricalcium phosphate with or without simvastatin [28], or the use of a bovine bone substitute with or without simvastatin [29], in maxillary sinus augmentation. In the first [28], radiographic follow-up was complemented with a biopsy after 9 months. Histomorphometric results showed that the amount of newly formed bone was statistically significantly higher in the simvastatin group. Because the patients of the intervention group showed an intense postoperative inflammatory reaction, the authors emphasize that the dose of simvastatin should be re-evaluated. The second pilot study [29] performed a radiographic follow-up until 9 months, evaluating alveolar bone height, the vertical height of the grafted bone and density. The results did not show any significant positive effect for simvastatin in maxillary sinus augmentation based on radiographic examination. Perhaps the histological study could have shown positive results, as the previous paper.

No other reviews evaluating the effect of local application of statins on the regeneration of non-periodontal bone defects have been found, but a recent systematic review on the use of statins in implantology for animal models (rats and dogs), obtained a significant increase in the osteogenesis around the implants, in cases where the drug was administered locally, applied directly to the surface of the implant [30]. In a similar way to the present work, the articles included in this review also mainly used simvastatin in different concentrations, although in some cases fluvastatin was used, and the follow-up periods ranged from 14 to 84 days.

There are meta-analyses that refer to the use of statins as an adjunct to scaling and root planning in humans. As two of the most recent analyses, it was concluded that the application of statins together with mechanical periodontal treatment significantly reduces the clinical attachment level and periodontal bone defects [31,32]. Both analyzed studies with long follow-up periods of up to 9 months, in which the statins used were simvastatin, rosuvastatin and atorvastatin, the latter not used in any of the studies of the present review. Although most of the studies that analyze statins used gels with concentrations of 1.2% and 2%, some work in which the drug is administered orally exist for both cases.

As far as limitations of this review go, it should be noted that the studies are different in terms of filler material (statin used, dose and complementary substances), the animal model and the parameters analyzed. In addition, the samples are small and the follow-up periods are short. Because some quantitative data are not provided in the studies, only two studies have been included in the meta-analyses.

Even the positive results obtained in animal model, considering the scarce and incipient pilot studies with ambiguous results, more homogeneous human studies, with larger and randomized samples and histological evaluation, are needed.

In conclusion, the local application of statins could be a promising therapeutic strategy for the regeneration of bone defects due to its apparent osteogenic and angiogenic effect. Further randomized clinical trials with larger samples and histological studies are necessary to establish its effect.

## Figures and Tables

**Figure 1 materials-12-02992-f001:**
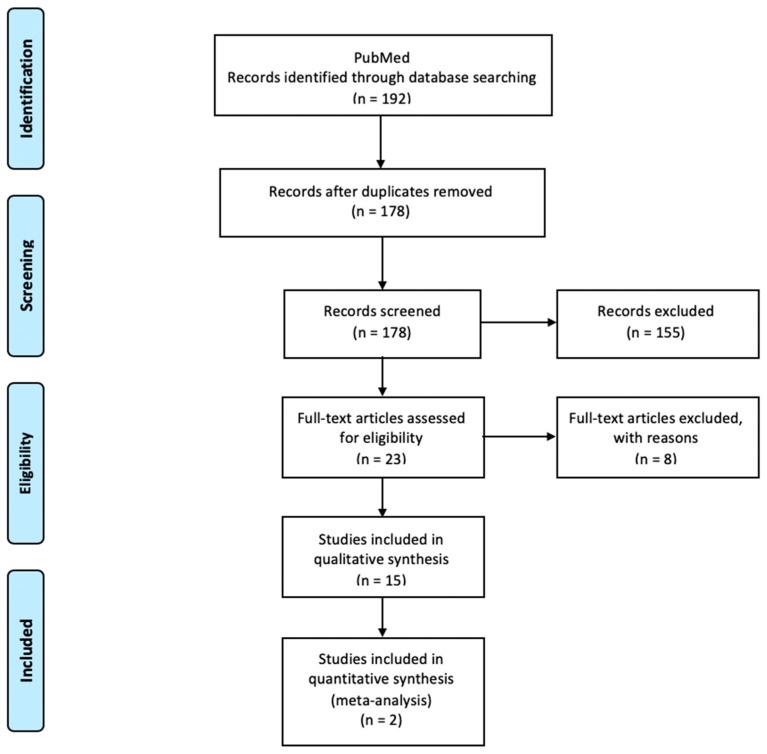
Preferred Reporting Items for Systematic Reviews and Meta-Analyses (PRISMA) flow diagram of selection process.

**Figure 2 materials-12-02992-f002:**
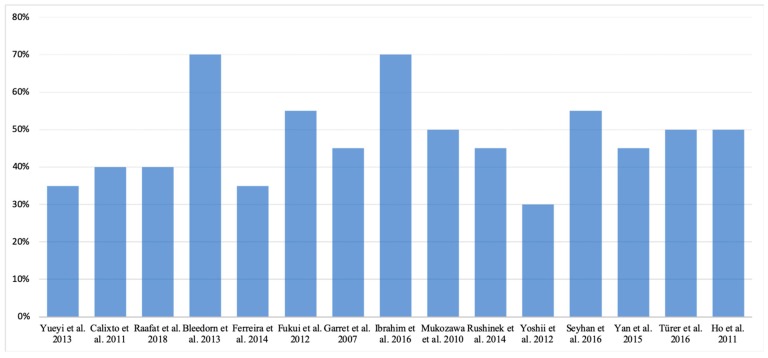
Quality assessment of the studies included according to the Animal Research: Reporting in vivo Experiments (ARRIVE) guidelines’ checklist.

**Figure 3 materials-12-02992-f003:**
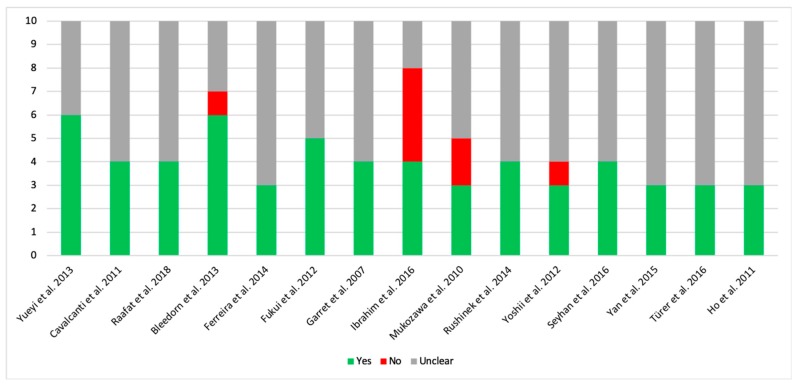
Risk of bias measured with the SYRCLE’s tool.

**Figure 4 materials-12-02992-f004:**
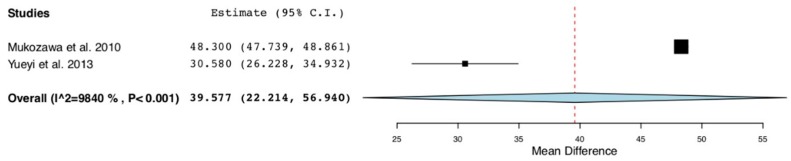
Forest plot: studies evaluating the percentage of bone formation at 4-6 weeks after the local application of statins in the treatment group versus no introduction of filling material or the introduction of polylactic acid in the control group.

**Figure 5 materials-12-02992-f005:**
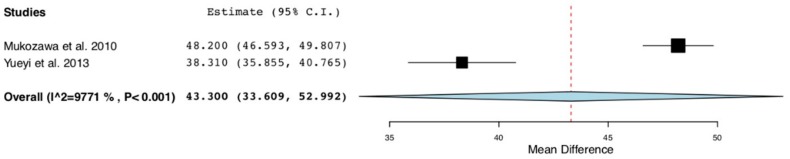
Forest plot: studies evaluating the percentage of bone formation at 12 weeks after the local application of statins in the treatment group versus no introduction of filling material or the introduction of polylactic acid in the control group.

**Table 1 materials-12-02992-t001:** Summary of the animal studies reviewed.

	Animal Species	N/Randomization	Filling Material	Follow-Up	Values Registered	Significant Results
**Yueyi et al. 2013** [8]	Rabbit and rat	16 and 8 NE	200 mg polylatcic acid (1a)// 200 mg polylatcic acid + 50 mg simvastatin (2a) (rabbit) // 20 mg polylactic acid (1b)// 20 mg polylactic acid + 5 mg simvastatin (2b) (rat)	42 and 72 days	New bone formation, BMP-2, HIF-1α, GFP-labeled BMSCs, EPCs and BMSCs	Bone formation, EPCs periphereal blood, BMSCs periphereal blood, GFP-labeled BMSCs, HIF-1α, BMP-2
**Calixto et al. 2011** [9]	Rat	64 NE	No filling material (1)// collagen sponges soaked in water (2)// collagen sponges + 2.2 mg/50 μL simvastatin (3)// collagen sponges + 0.5 mg/50 μL simvastatin (4)	30 and 60 days	BMD, histomorphometry	Group 3 radiographic density at 30 days
**Raafat et al. 2018** [10]	Rat	48 NE	No filling material (1)// 6 mg PRF(2)// 1mg simvastatina + gelatin (3)// 1mg simvastatin + PRF 1:6 (4)	30 and 60 days	New bone formation, histomorphometry, BMP-2, VEGF; OPG, RANKL, ALP, OSC and BMD	Bone formation in groups 2, 3 and 4; bone maduration in group 4 at 60 days; BMP-2 and VEGF in groups 2, 3 and 4; OPG, OSC and ALP in groups 3 and 4; RANKL in groups 3 and 4; BMD in group 4; complete bone healing at 60 days in group 4
**Bleedorn et al. 2013** [11]	Dog	18 Randomized	25% polyethylene glycol 400 + 75% hyaluronic acid (1)// 150 mg lovastatin + polyethylene glycol 400 + 75% hyaluronic acid (2)	70 days (77 in 2 cases)	Radiographic bone union, time bone healing, histomorphometry	Bone healing, soft tissue necrosis and inflammation
**Ferreira et al. 2014** [12]	Rat	66 NE	No filling material (1)// no filling material + poly(lactic-co-glycolic acid) membrane (2)// 5 mg simvastatin + poly(lactic-co-glycolic acid) microspheres + poly(lactic-co-glycolic acid) membrane (3)// poly(lactic-co-glycolic acid) microspheres + poly(lactic-co-glycolic acid) membrane (4)	30 and 60 days	New bone formation, OPN, BSP, OSAD, histomorphometry	Bone healing, bone matrix organization and maturity
**Fukui et al. 2012** [13]	Rat	60 NE	250 mg simvastatin conjugated with gelatin hydrogel (1)// gelatin hydrogel alone (2)	14, 28 and 56 days	Bone healing, histomorphometry, RT-PCR analysis, capillary density and OB density, blood perfusion, biomechanical analysis (stress, extrinsic stiffness, failure energy, RR of fractured femur to nonfractured, EPCs	Bone healing, angiogenesis, osteogenesis, bone density
**Garret et al. 2007** [14]	Rat	72 NE	50 μL PBS (1)// 50 μL biodegradable polymer nanoparticle (2)// 50 μL biodegradable polymer nanoparticle delivering 0.2 μg/day lovastatin (3)// 50 μL biodegradable polymer nanoparticle delivering 1μg/day lovastatin (4)// 50 μL biodegradable polymer nanoparticle delivering 1.5 μg/day lovastatin(5)// 50 μL biodegradable polymer nanoparticle delivering 7.5 μg/day lovastatin (6)	14 and 28 days	Bone healing, biomechanical measurements, lovastatin plasma levels	Bone healing, cortical fracture gap at 4 weeks, maximum force to fracture and work-to-fracture (groups 4 and 5), lovastatin undetectable at 28 days
**Ibrahim and Fahmy 2016** [15]	Rat	16 NE	3:1 chitosan to Carbapol^®^ + 2% Imwitor^®^ + 19.88-24.38 mg rosuvastatin sponges (1)// no filling material (2)	30 days	Bone healing	Bone healing
**Mukozawa et al. 2010** [16]	Rabbit	20 NE	2.5 mg/mL simvastatin in 0.2 ml water + hydrogel (1)// 2.5 mg/mL simvastatin in 0.2 ml water + atelocollagen sponge (2)// hydrogel (3)// atelocollagen sponge (4)// no filling material (5)	7, 14, 28, 56 and 84 days	New bone area ratio, BMP-2, histomorphometry	Number cells stained positive to BMP-2 at 14 and 28 days (groups 1 and 2), new bone area ratio at 14, 28, 56 and 84 days (groups 1 and 2)
**Rushinek et al. 2014** [17]	Rat	16 NE	Slow-release degradable hydroxypropyl methylcellulose (70% simvastatin and 30% Methocel K100M) (1)// Slow-release degradable hydroxypropyl methylcellulose (100% Methocel K100M) (2)	14, 28, 42 and 56 days	MBV/TV, Ob.S/BS, OS/BS, OV/BV, Os.Th, MS/BS, MAR, BFR/BS, double-labeled calcein surface	Double-labeled surface, OV/BV, Os.Th
**Yoshii et al. 2012** [18]	Rat	18 and 18 NE	Polyethylene glycol in PBS (200 μL) (1a)// Polyethylene glycol in PBS (200 μL) + 25 μg LV-MPs (2a)// Polyethylene glycol in PBS (200 μL) + 100 μg LV-MPs (3a)// Polyethylene glycol in PBS (200 μL) at 14, 28 and 42 days (1b)// Polyethylene glycol in PBS (200 μL) + 100 μg LV-MPs at 14, 28 and 42 days (2b)	28 days/ 14, 28, 42 and 56 days	Mineralized bone formation, bone volume, density in the defects, newly formed bone matrix	Volume and density of newly formed bone (3a)// Volume and density of newly formed bone at 56 days (2b)
**Seyhan et al. 2016** [19]	Rat	30 Randomized	PBS (1)// 0.5 mL PRP (2)// 0.1 mL simvastatin (3)	56 and 112 days	New bone forming area, fibroblasts, osteoblasts, osteoclasts, vessel diameter	New bone forming area, fibroblasts, osteoblasts, vessel diameter and osteoclasts (3)
**Yan et al. 2015** [20]	Rat	24 Randomized	PLGA-PEG-PLGA (1)// SIM/PLGA-PEG-PLGA(2)// no filling material(3)	28 days	New bone formation ratio, histomorphometry	New bone formation ratio (2)
**Türer et al. 2016** [21]	Rat	32 NE	Collagen sponge (1,2) // Collagen sponge with saline solution containing 1 mg rosuvastatin (3,4)	14 and 28 days	New bone volume	New bone volume at 14 days (3,4)
**Ho et al. 2011** [22]	Rat	30 Randomized	1 mg PLGA nanoparticles containing lovastatin (1a) // gelfoam (1b) // 3 mg nanoparticles containing lovastatin (2a) // gelfoam (2b) // 1 mg nanoparticles containing lovastatin (3a) // 1 mg nanoparticles without lovastatin (3b) // 3 mg nanoparticles containing lovastatin (4a) // 3 mg nanoparticles without lovastatin (4b)	21, 42, 63 and 84 days	Volume changes of the defect, histomorphometry	Remaining bony defect in volumen (1a) at 42, 63 and 84 days

Abbreviations: ALP: alkaline phosphatase enzyme, BFR/BS: bone formation rate, BMD: bone mineral density, BMP-2: bone morphogenetic protein-2, BMSC: bone marrow-derived mesenchymal stem cell, BSP: bone sialoprotein, EPC: endothelial progenitor cell, GFP: green fluorescent protein, HIF-1α: hypoxia-inducible transcription factor 1-alpha, LV-MP: injectable lovastatin microparticles, MAR: mineral apposition rate, MBV/TV: mineralized bone volume, MS/BS: mineralized surface, NE: not specified, OB: osteoblasts, Ob.S/BS: osteoblast surface, OPG: osteoprotegerin, OPN: osteopontin, OSAD: osteoadherin, OS/BS: osteoid surface, OSC: osteocalcin, Os/Th: osteoid thickness, OV/BV: osteoid volume, PBS: phosphate-buffered saline, PLGA: poly(lactic-co-glycolic acid), PLGA-PEG-PLGA: poly (D,L-lactide-co-glycolide)-poly(ethylene glycol)-poly (D,L-lac-tide-co-glycolide), PRF: platelets rich fibrin, PRP: platelet-rich plasma, RANKL: receptor activator of nuclear factor kappa-B ligand, RR: relative ratio, RT-PCR: real time polymerase chain reaction, SIM/PLGA-PEG-PLGA: simvastatin/poly(D,L-lactide-co-glycolide)-poly(ethylene glycol)-poly (D,L-lac-tide-co-glycolide), VEGF: vascular endothelial growth factor.

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
