# Peer review of "The Application of Statins in the Regeneration of Bone Defects. Systematic Review and Meta-Analysis"

_materials, 2019, doi:10.3390/ma12182992_

Round 1

Reviewer 1 Report

I believe that the manuscript is suitable for publication, and would only present the data both in the abstract and in the text of the manuscript with a single decimal.

Author Response

Reviewer #1

I believe that the manuscript is suitable for publication, and would only present the data both in the abstract and in the text of the manuscript with a single decimal.

# Thank you very much for your assessment. We have made the changes suggested to us (In red in the manuscript)

Reviewer 2 Report

The manuscript has been revised, but it still needs some more corrections.  I believe the authors should take into account following suggestions which were not sufficiently answered:

I am not familiar with the Cochrane Scielo database,  I would like some information about it. Actually I had expected the authors to perform a further search in Embase or Cochrane Library. The keywords as well as meshterms used are still not enough. I suggest, a more detailed combination of terms such as (HMG-CoA reductase inhibitors OR lipid-lowering medications) AND (bone healing OR bone turnover OR bone metabolism) should be performed leading to a more thorough literature search.

Author Response

Reviewer #2

The manuscript has been revised, but it still needs some more corrections. I believe the authors should take into account following suggestions which were not sufficiently answered:

I am not familiar with the Cochrane Scielo database, I would like some information about it. Actually I had expected the authors to perform a further search in Embase or Cochrane Library. The keywords as well as meshterms used are still not enough. I suggest, a more detailed combination of terms such as (HMG-CoA reductase inhibitors OR lipid-lowering medications) AND (bone healing OR bone turnover OR bone metabolism) should be performed leading to a more thorough literature search.

# Thank you very much for your assessment. Scielo is a database that allows access to articles published by open access journals belonging to South America, the Caribbean, Spain, Portugal and South Africa. After reading the comments, we have expanded the search using the suggested keywords (HMG-CoA reductase inhibitors OR lipid-lowering medications) AND (bone healing OR bone turnover OR bone metabolism) in Pubmed/Medline, Embase, Cochrane Library and Scielo databases. We have identified 10 possible articles to include, which after applying the inclusion and exclusion criteria established in our work, have resulted in 4 articles that we have included in our review. Thus, we have modified the results taking into account these four studies, as well as the table and graphs. Unfortunately, the absence of data in these articles has not allowed us to include them in the fores-plot. 

(In red in the manuscript)

Reviewer 3 Report

This is an excellent review and meta-analysis on the effect of the local application of statins in the regeneration of non-periodontal bone defects. The authors found out that local application of the drug increased the percentage of the following parameters: new bone formation, bone density, bone healing, bone morphogenetic protein 2, vascular endothelial growth factor, progenitor endothelial cells and osteocalcin. They conclude that local application of statins could be a promising therapy aimed at regeneration of bone defects. The effects is apparently mediated by both osteogenic and angiogenic effect. Due to the very scarce literaure in humans, they also warn us that further randomized clinical trials with larger samples and histological studies are necessary to establish its effect.

The paper is clearly written. The methodolgy is sound and the conclusions are supported by the results.

Author Response

Reviewer #3

This is an excellent review and meta-analysis on the effect of the local application of statins in the regeneration of non-periodontal bone defects. The authors found out that local application of the drug increased the percentage of the following parameters: new bone formation, bone density, bone healing, bone morphogenetic protein 2, vascular endothelial growth factor, progenitor endothelial cells and osteocalcin. They conclude that local application of statins could be a promising therapy aimed at regeneration of bone defects. The effects is apparently mediated by both osteogenic and angiogenic effect. Due to the very scarce literaure in humans, they also warn us that further randomized clinical trials with larger samples and histological studies are necessary to establish its effect.

#Thank you very much for your comments and for taking the interest to review the manuscript.

Reviewer 4 Report

No comments

Author Response

Reviewer #4

No comments

#Thank you very much for reviewing the manuscript.

This manuscript is a resubmission of an earlier submission. The following is a list of the peer review reports and author responses from that submission.

Round 1

Reviewer 1 Report

This is a very good meta analysis . The methods are clearly described and the conclusions are well supported by the results (i.e. animal studies support the notion that  local application of statins promotes healing of critical bone size defects due to its apparent osteogenic and angiogenic effect.  . I recommend to publish after a minor editing  for English language

Reviewer 2 Report

The manuscript entitled "The application of statins in regeneration of bone 2 defects. Systematic review and meta-analysis" presents a correct justification and addresses an interesting topic. Although the authors declare that they have followed the PRISMA recommendations for carrying out this work, some important points have been omitted:

-only have used a database (medline / pubmed). It is necessary to use at least 3 different databases. I do not understand how they find 14 duplicates in the search.

-The meta-analysis is very risky, with only 2 studies included and with a high heterogeneity (although convergent) apparently due to the difference existing in the comparison groups of the two studies (no introduction of filling material or introduction of polylactic 28 acid).

-The publication bias has not been analyzed, due to the small sample size (only two studies included).

Although the conclusions are adequate, and some methodological aspects such as the evaluation of the quality of the studies are rightly performed, I consider that there is a high probability that if the authors extend the search in other databases such as EMBASE, SCOPUS or WOS they could include more studies that would reinforce the evidence obtained. 

Therefore, I would recommend to authors expanding the search in order to be publishable in impact dental journals.

Reviewer 3 Report

This systematic review shows significant methodological weaknesses and inconsistencies.

There is a lack in the methodology, and a poor systematic experimental design. There is no focused question given in the Materials and Methods section. The electronic literature search strategy is inadequately presented, very few mesh terms (only 3 key words) are used in the electronic search, probably leading to an insufficient retrieve of relevant publications. For example one relevant article reporting on the same subject was not referred:

"Effects of local application of simvastatin on bone regeneration in femoral bone defects in rabbit."  Papadimitriou et al. 2015, Journal of Cranio-maxillo-facial surgery, 43, 2015, 232-237.

 I would like to ask the authors, why they did not include the above article in their study.

Furthermore the forest plot comparisons performed in the Results section are inappropriate. The authors are analysing bone formation outcome assessed by non homogeneous methodology in the included studies in their meta- analysis, One study (Mukozawa 2011) is using histomorphometric evaluation of BMP-2 while the other (Yueyi 2013) is using radiographic (CT) measurements. Accordingly the meta-analysis is irrelevant and should be disregarded.